# Business Email Compromise (BEC) Attacks: Threats, Vulnerabilities and Countermeasures—A Perspective on the Greek Landscape

Anastasios Papathanasiou [1,2,*] , George Liontos [3] , Vasiliki Liagkou [2,*] and Euripidis Glavas [2]

1   Cyber Crime Division, Hellenic Police, 173 Alexandras Avenue, 11522 Athens, Greece
2   Department of Informatics and Telecommunications, University of Ioannina, Kostaki Artas, 47150 Arta, Greece; eglavas@uoi.gr
3   Department of Materials Science and Engineering, University of Ioannina, 45110 Ioannina, Greece; gliontos1985@gmail.com
*   Correspondence: anastasios.papathanasiou@gmail.com (A.P.); liagkou@uoi.gr (V.L.)

**Abstract:** Business Email Compromise (BEC) attacks have emerged as serious threats to organizations in recent years, exploiting social engineering and malware to dupe victims into divulging confidential information and executing fraudulent transactions. This paper provides a comprehensive review of BEC attacks, including their principles, techniques, and impacts on enterprises. In light of the rising tide of BEC attacks globally and their significant financial impact on business, it is crucial to understand their modus operandi and adopt proactive measures to protect sensitive information and prevent financial losses. This study offers valuable recommendations and insights for organizations seeking to enhance their cybersecurity posture and mitigate the risks associated with BEC attacks. Moreover, we analyze the Greek landscape of cyberattacks, focusing on the existing regulatory framework and the measures taken to prevent and respond to cybercrime in accordance with the NIS Directives of the EU. By examining the Greek landscape, we gain insights into the effectiveness of countermeasures in this region, as well as the challenges and opportunities for improving cybersecurity practices.

**Keywords:** Business Email Compromise (BEC); cybercrime; social engineering; phishing; Greece legislation NIS compliance

## 1. Introduction

As we live in an age of rapid technological advances, and as people, companies and organizations rely more and more on services like online communication, cloud computing, social networks and online fund transfers, the space for malicious cyberattacks has increased dramatically [1]. Email, due to its high usage across the world as a mean of communication and as a work tool, is used to spread malware, spam and phishing attacks. Phishing is a type of cyberattack in which a person/actor sends messages pretending to be a trusted person or entity to try to lure people with deceptive content to share sensitive information, click on a malicious URL, download malicious attachments, or—in most cases—request in an urgent manor that they make a financial transaction. Phishing in the form of a well-crafted email and conducted through gathering and spoofing information about the target has proven to be a very lucrative scam called Business Email Compromise (BEC). The outbreak of COVID-19 resulted in physical lockdowns, an increase in working from home, remote access to business resources, and online shopping and online payments, thus leading to individuals and business becoming more vulnerable to cybercrime activities such as phishing, smishing, ransomware, malware, Business Email Compromise (BEC) attacks etc. [2]. COVID-19 not only caused the volume of cyberattacks to increase, but also favored attacks based on social engineering [3]. Vishing, smishing and BEC attacks, in

combination with spoofing, whereby actors use legitimate-looking IDs, text aliases and mutated email addresses, saw a significant increase in number and credibility, according to the Internet Organized Crime Threat Assessment (IOCTA) [4]. At this point, it is worth mentioning that the skills required to enter into cyber criminality are becoming lower and lower. Hacking tools nowadays are commonly available even outside the dark web, often with the disguise of ethical hacking tools, accompanied by video guides and step by step tutorials. The most notable examples of all are the cases of Kali Linux and Parrot OS. Guided by YouTube instructions, someone with limited knowledge of cybersecurity can install a virtual workstation and then Kali Linux or Parrot OS, which both come with preinstalled tools for:

- Information Gathering (nmap, legion, maltego etc.);
- Vulnerability Analysis (nmap, legion, nikto etc.);
- Web Application Analysis (sqlmap, wpscan, ZAP etc.);
- Database Assessment (sqlmap, SQLite etc.);
- Passwords Attacks (ncrack, medusa, hashcat etc.);
- Wireless Attacks (wifite, aircarck, reaver etc.);
- Reverse Engineering (clang, radare2, NASM shell etc.);
- Exploitation Tools (Metasploit framework, msf payload creator, social engineering toolkit etc.);
- Sniffing and Spoofing (macchanger, wireshark, responder etc.);
- Post Exploitation (mimikatz, weevely, powesploit etc.);
- Social Engineering Tools (maltego, msf payload creator, social engineering toolkit etc.);

These tools are meant to be used for ethical hacking and for detecting possible system exploitations, but the line between ethical and unethical hacking, as history has shown, is very thin. With those tools provided and information and malware resources from the dark web, someone with originally limited knowledge of cybersecurity can perform all kinds of malicious actions. On the other hand, despite the fact that cyberattacks (especially malware, ransomware and BEC) are a major threat to enterprises scaling from small (1–50 employees), to medium (50–250 employees), to large (1000+ employees), in many cases, cybersecurity issues and measures need to be taken but are not a priority boardroom discussions. In fact, as Kasperksy's report ([5]) highlights, the bigger the company is, the less important cybersecurity is at the board level. While C-suite executives recognize security attacks as the number one risk businesses face, just over half (51%) stated that cybersecurity is always an agenda item for their board meetings (see Figure 1).

| | UK | France | DACH | Benelux | Spain | Portugal | Italy | Romania | Greece |
|---|---|---|---|---|---|---|---|---|---|
| Cybersecurity attacks | 57.0% | 46.0% | 61.0% | 52.0% | 45.5% | 51.5% | 44.0% | 45.0% | 43.0% |
| Economic factors | 30.5% | 37.0% | 35.0% | 44.0% | 40.5% | 33.0% | 41.0% | 45.5% | 23.0% |
| Regulation / compliance | 27.0% | 36.0% | 35.0% | 35.5% | 38.5% | 35.0% | 34.5% | 37.0% | 34.0% |
| Natural disasters | 26.0% | 36.5% | 29.0% | 30.0% | 40.5% | 26.5% | 31.0% | 32.5% | 26.0% |
| Competitors | 30.5% | 30.0% | 26.5% | 30.0% | 31.5% | 30.5% | 28.0% | 25.0% | 31.0% |
| Environmental issues | 26.0% | 31.5% | 25.0% | 32.0% | 37.0% | 20.0% | 29.5% | 29.0% | 28.0% |
| Industrial action | 29.5% | 30.0% | 29.0% | 23.0% | 27.5% | 29.5% | 26.0% | 26.0% | 34.0% |

**Figure 1.** Biggest risks/threats facing business continuity [5].

*Contribution*

Mitigating the threat of Business Email Compromise (BEC) attacks requires ongoing efforts from organizations and individuals. Attackers constantly evolve their tactics, employing sophisticated phishing techniques, social engineering, and email spoofing to deceive victims. To effectively counter these attacks, cybersecurity awareness and comprehensive defense strategies are essential. In our study, we aimed to provide insights into BEC attacks, delving into the complexity of the scheme and the social engineering techniques that are employed. We also aimed to offer practical guidance for individuals and companies to protect themselves against BEC attacks, and we informed them about the actions needed after a BEC attack to minimize the consequences of the scheme and to facilitate the work of the authorities. By exploring social engineering tactics, we aimed to enhance the understanding of psychological manipulation techniques used by attackers, empowering individuals and organizations to recognize and respond effectively to BEC attacks. Proactive defense measures were proposed to mitigate the risk of falling victim to deceptive schemes. In addition, we performed an extensive analysis of Greece, chosen as a representative case within the European Union, to illustrate its efforts in combating cybercrime through legislative measures and the involvement of relevant departments and authorities. The significance of a country's legislation and defense strategies against cybercrime extends beyond its own borders, as they directly impact the security of individuals and enterprises operating within its jurisdiction. Overall, the contribution of this study is that it provides all the necessary information about BEC schemes in coordination with the existing European legislation. The study also suggests various measures for preventing and countering BEC schemes, and it also recommends actions needed to be taken after the attack takes place. Legislation is a critical factor that is often overlooked and contributes not only to defining the general concept (such as penalties and fines) of illegal actions, but in this case, regarding cybercrime, enforces various technical and non-technical measurements in order for individuals, enterprises and critical infrastructures to be efficiently protected. Figure 2 exhibits the main sections of the article and aims to inform the reader about BEC attacks and how to counter them.

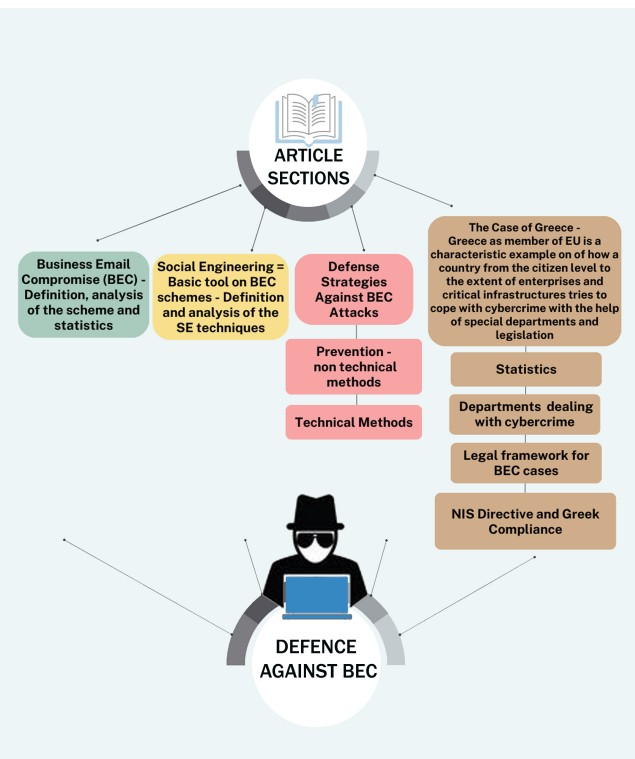

**Figure 2.** Main sections of this study.

## 2. Research Methodology

To identify relevant studies and gather suitable materials for our research, we employed various procedures for each major section of this paper. For the Section 1, we obtained statistics on cybercrime activities, losses, incidents, and other relevant facts from specialized divisions responsible for cybersecurity, such as Europol, the FBI, and ENISA. We also relied on reports from top-notch cybersecurity companies that specialize in this field. To explore the concept of the Business Email Compromise (BEC) scheme and Social Engineering, we conducted searches using keywords like "Business email compromise", "Detection of BEC schemes", "Social engineering", "cybercrime", "machine learning", and others. We utilized academic search engines like Google Scholar, Core, Scopus and Science.gov for this purpose. Following a pilot search, we employed an inclusion/exclusion procedure where articles irrelevant to our study were excluded, while those relevant to our research were included and analyzed. Furthermore, additional searches using the referenced works of relevant articles were also conducted (snowball effect). For Section 2 of the article, we conducted searches using keywords such as "Greek cybercrime", "Greek departments for cybersecurity", "NIS1/NIS2/Directive 2022/2555", "Joined Investigation Teams", "European legislation for cybersecurity" and others, using the same academic search engines mentioned earlier. Again, we followed an inclusion/exclusion procedure. However, there was a slight difference in this section as the work of the Maglaras team on the NIS 1 directive and Greece's compliance with NIS 1 were already known to the authors. This information served as a guide for analyzing the harmonization of Greece with all the directives. By employing these comprehensive and systematic procedures, we ensured that our research included relevant studies and materials, enhancing the depth and analysis of our paper. Although this methodology has been employed in several systematic literature studies, it possesses certain limitations. One limitation is that it may narrow the focus of the review or study, potentially leaving readers without a comprehensive understanding of the subject matter. Additionally, our data collection was limited to only four scientific sources, which may have restricted the number of publications included in our review. While these sources are recognized as reliable, the limitation arises from not exploring all potential sources to identify relevant articles concerning our study objectives. Furthermore, our examination of Greece as a case study for a European country's efforts to combat cybercrime through European legislation acknowledges that the implementation and harmonization of such measures may vary among EU member states due to various economic, social or other factors beyond the scope of this paper. Lastly, although we present a range of non-technical and technical methods for individuals and enterprises to combat BEC attacks in conjunction with the legal framework provided by the EU, we acknowledge that alternative technical solutions may exist, particularly in the field of machine learning, which we did not extensively cover in this paper to maintain focus and coherence.

## 3. Literary Review

Business Email Compromise (BEC), as a man-in-the-middle email attack, is a type of cybercrime where a scammer uses sophisticated crafted emails or computer intrusion techniques in order to trick someone into conducting unauthorized transfers of funds or divulging confidential company information. The attacker usually starts by researching the profile of the employees, gathering information from online data and social media, and monitoring ongoing conversations in the victim's organization. This information is crucial in order for a highly convincing email to be crafted. BEC mails are targeted, and the body text is usually crafted in a familiar language context for the victim. Also, BEC mails are sent from compromised accounts, spoofed addresses or seemingly legitimate domains and so they cannot be detected from spam filters. BEC attacks usually come in the form of:

- Spear Phishing attack: in which a hacker targets a specific individual or organization, undergoing a thorough research of the target's information to increase the chances of the scam attempt being successful.

- Whaling attack: a type of spear phishing attack, but in this case, the target is usually a high-profile executive such as a CEO or CFO [6].

According to the FBI's 2022 report [7] BEC is one of the fastest growing, most financially damaging internet-enabled crimes, and is a major threat to the global economy. The sophistication of BEC criminal actors and their ever-evolving tactics has increased over time, likely driving increased dollar losses. BEC actors have targeted large and small companies in every U.S. state and more than 150 countries around the world. As IC3 mentions [8] in its 2022 crime report, the agency reports 21,832 Business Email Compromise (BEC)/Email Account Compromise (EAC) victims, with adjusted losses at nearly USD 2.7 billion. Figure 3 shows a chart from IC3 with the numbers of victims for various crime types for 2022, Figure 4 shows an estimation of victim losses categorized by crime type for 2022, and Figure 5 shows Greece's place in a chart with the top 20 countries by number of total victims as compared to the United States for 2022.

| Crime Type | Victims |
|---|---|
| Phishing | 300,497 |
| Personal Data Breach | 58,859 |
| Non - Payment /Non - Delivery | 51,679 |
| Extortion | 39,416 |
| Tech Support | 32,538 |
| Investment | 30,529 |
| Identity Theft | 27,922 |
| Credit Card / Check Fraud | 22,985 |
| BEC | 21,832 |
| Spoofing | 20,649 |
| Confidence / Romance | 19,021 |
| Employment | 14,946 |
| Harassment / Stalking | 11,779 |
| Real Estate | 11,727 |
| Government Impersonation | 11,554 |
| Advanced Fee | 11,264 |
| Other | 9,966 |
| Overpayment | 6,183 |
| Lottery / Sweepstakes / Inheritance | 5,65 |
| Data Breach | 2,795 |
| Crimes Against Children | 2,587 |
| Ransomware | 2,385 |
| Threars of Violence | 2,224 |
| IPR / Copyright / Counterfeit | 2,183 |
| SIM Swap | 2,026 |
| Malware | 765 |
| Botnet | 568 |

**Figure 3.** Chart with number of victims for various crime types [8].

| By Victim Loss | | | |
|---|---|---|---|
| Crime Type | Loss | Crime Type | Loss |
| Investment | $3,311,742,206 | Lottery/Sweepsthakes/Inheritance | $83,602,376 |
| BEC | $2,742,354,049 | SIM Swap | $72,652,571 |
| Tech Support | $806,551,993 | Extortion | $54,335,128 |
| Personal Data Breach | $742,438,136 | Employment | $52,204,269 |
| Confidence/Romance | $735,882,192 | Phishing | $52,089,159 |
| Data Breach | $459,321,859 | Overpayment | $38,335,772 |
| Real Estate | $396,932,821 | Ransomware | $34,353,237 |
| Non-Payment/Non-Delivery | $281,770,073 | Botnet | $17,099,378 |
| Credit Card/Check Fraud | $264,148,905 | Malware | $9,326,482 |
| Government Impersonation | $240,553,091 | Harassment/Stalking | $5,621,402 |
| Identity Theft | $189,205,793 | Threats of Violence | $4,972,099 |
| Other | $117,686,789 | IPR/Copyright/Counterfeit | $4,591,177 |
| Spoofing | $107,926,252 | Crimes Against Children | $577,464 |
| Advanced Fee | $104,325,444 | | |

**Figure 4.** Estimation of victim loss categorized by crime type for 2022 [8].

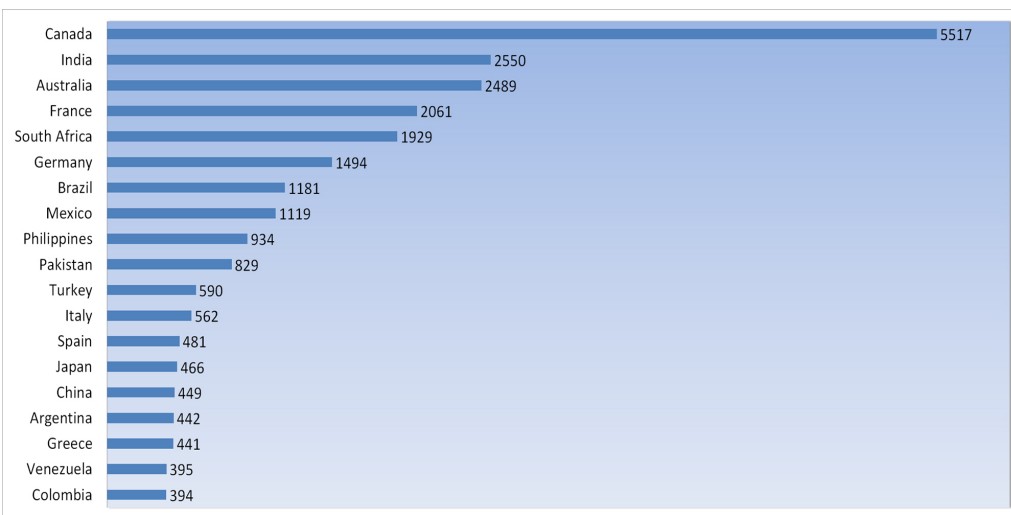

**Figure 5.** Chart of the top 20 countries by number of total victims as compared to the United States for 2022. The specific number of victims for each country are listed in ascending order to the right of the graph [8].

The European Union Agency for Cybersecurity (ENISA) also reports that BEC is one of the most impactful types of cybercrime. Compared to previous years, the median transaction size for Business Email Compromise attacks further increased substantially, despite the efforts of law enforcement agencies to suppress this tension [9]. Figure 6 shows a chart with a time series of major incidents observed by ENISA (July 2021–June 2022) concerning social engineering threats.

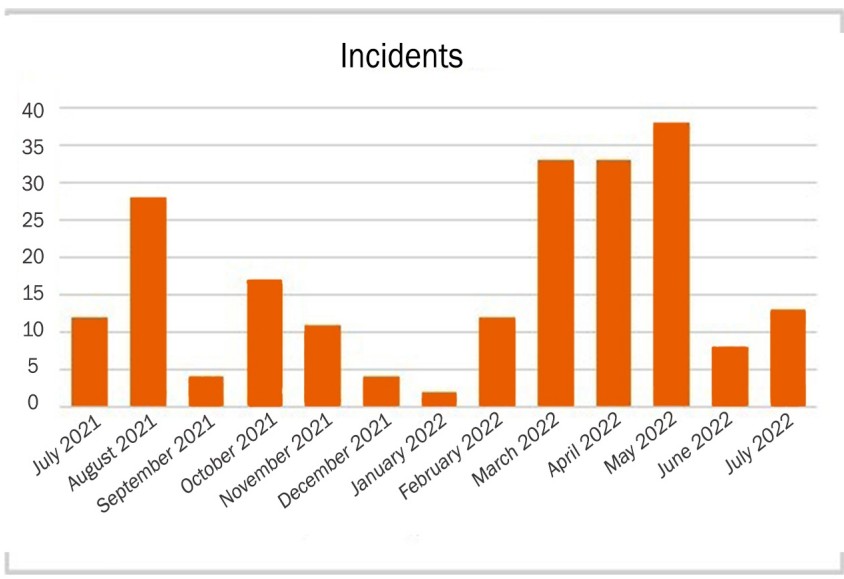

**Figure 6.** Time series of major incidents observed by ENISA (July 2021–June 2022) concerning social engineering threats [9].

Data Breach Investigation (DBIR) reports for 2022 say that only 41% of BECs involved phishing. Of the remaining 59%, 43% involved the use of stolen credentials against the victim organization. The percentage remaining were most likely BECs using an email from a partner. Figure 7 shows the median transaction size for BECs base on FBI IC3 complaints where a transaction occurred. It seems that scammers in 2022 increased their demands [10].

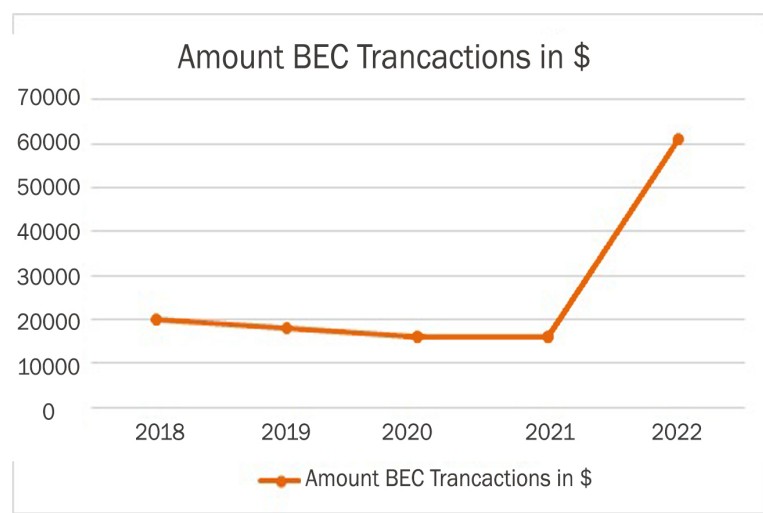

**Figure 7.** Transaction size for BECs base on FBI IC3 complaints where a transaction occurred [10].

According to the Cloudflare 2021 security report [11], BECs made up only 1.3% of attacks and an even smaller percentage of total email volume. However, because of the fact that a BEC attack is extremely targeted due to the extreme research of the target's profile, the financial cost of this type of attack is disproportionate to the above mentioned volume. Table 1 shows some of the costliest BEC attacks hitting major brands.

**Table 1.** Some of the costliest BEC attacks on major brands according to Cloudflare 2022 Security Report [12].

| Victim | *Loss* | Year | What Happened |
|---|---|---|---|
| Facebook and Google | *$123 Million* | 2019 | A Lithuanian scammer *impersonated* Quanta Computer, an electronics supplier for Facebook and Google. Facebook and Google paid $123 million in fake invoices to the scammer. |
| Crelan Bank | *$75.8 Million* | 2016 | Belgian-based Crelan Bank lost over €70 million (roughly $75.8 milllion) to fraudsters who compromised the CEO's email account. The *attack* was later discovered in an internal audit. |
| Toyota Subsidiary | *$37 Million* | 2019 | A European subsidiary of the Toyota Boshoku Corporation(subsidiary of the Toyota Group) was *duped* into transferring approximately $37 billion in a BEC scam |
| Scoular | *$17.2 Million* | 2014 | US commodities trading company Scolar *wired* $17.2 million to a fraudulent offshore account after receiving a fake email from the CEO. |
| Mattel | *$ 3 Milllion (recovered)* | 2016 | A financial executive at Mattel *transferred* $3 million to a fraudulent account after receiving a spoofed email appearing to be from the CEO. |

BEC attacks can be categorized in three major techniques according to Remorin et al. [13]:

- Phishing-related techniques: A typical phishing attack involves the use of email containing a malicious URL link, PDF attachments containing malicious code or pointing to URLs, HTML files, and file-hosting service sites.
- Malware-related technique: The use of malware for BEC attacks such as keyloggers and remote access tools (RATs) have proven extremely useful due to their effectiveness and low cost. This kind of malware is commonly available in hacking forums or

in the dark web and has evolved to use crypto services to evade detection from antivirus programs.

- Social engineering-based technique: This is the most common technique, where a well-crafted email is sent from the attacker who, through extensive research, is impersonating a entity known to the victim, usually a friend, associate, client, etc.

These are the common characteristics of emails used in a BEC attack:

1. Subject: Email subjects that contain the words "request", "payment", "urgent" and "wire transfer" are a red flag for a BEC attack.
2. Reply-To: Due to the fact that most email clients do not display the reply-to addresses, actors insert their email address on the reply-to portion of the mail.
3. From: A legitimate-looking email address is crafted to impersonate a person of trust.
4. Frequency: The attack is more likely to succeed if BEC actors have checked the location and the time zone of the target. In the case of an attacker impersonating a CEO requesting a wire transfer, sending a BEC email during working hours on Friday and demanding that the request has to be resolved before the weekend has proven particularly successful.

Jakobsson [14], on the other hand, reports three principal methods BEC scammers use to disguise themselves as a known entity to their victims:

- By gaining access to the victim's mail account, commonly referred to as Account Take-Over (ATO). Access is usually gained with phishing techniques or malware.
- By spoofing a trusted party. Attackers usually spoof email addresses or reply-to addresses. DMARC has outdated this method, but not all organizations support it.
- Through social engineering techniques and information from social networks like LinkedIn, a scammer can discover relationships between the victim and his/her associates and then register a deceptive domain name or email account. In this case, no spoofing was needed, meaning that DMARC cannot protect the victim from this kind of scam.

*The Role of Social Engineering in BEC Attacks*

BEC attacks rely heavily on social engineering. Social engineering is the art of getting users to compromise information systems and procedures. Instead of technical attacks on systems, social engineers target humans and, through influence and persuasion, mislead them into giving sensitive information, click on malicious URLs and attachments, or even into carrying out actions like unauthorized wire transfers [15]. Scammers usually prefer payment methods like wire transfers because they are easy to fabricate and easy to perform but difficult to reverse. Common targets of social engineering include company clients, help desk personnel, technical support executives, system administrators, financial executives, lawyers, CEO's, etc. In order to perform a successful social engineering attack, the attacker must gather information; this process is called OSINT (Open Source Intelligence). OSINT can be non-technical (observation skills) or technical, meaning that information can be gathered from sources like:

- Social media (LinkeIn, Facebook, Twitter, etc.);
- Search engines;
- Official web sites;
- Forums, blogs etc.;
- Advertisements;
- Services or commands like pipl, WHOIS lookup (Linux or websites), FOCA, Maltego;
- Metadata (documents, photos etc.).

With the use of the above mentioned gathered information, a social engineer can d0x a target. The word d0x is a hacking term that means to create a document containing details and information about the target [16]. Applying this information in combination with some

of the basic social engineering principles is the next step. According to [16–18], some basic social engineering principles are:

- Authority: This is a particularly effective technique because people are more likely to respond to authority. Attackers can claim authority with various ways like verbal communication, spoofing an email, etc.
- Intimidation: In many cases, intimidation derives from authority. Authority, confidence or even threats of harm can motivate a person to perform actions on an attacker's behalf.
- Consensus: The act of taking advantage of a person's natural tendency to mimic what others are doing or are perceived as having done in the past.
- Scarcity: A technique used to convince someone that an object has a higher value based on the object's scarcity. Scarcity can also be applied to time or information to increase the influence on the target.
- Familiarity, liking, similarity: A principle that exploits a person's native trust in that which is familiar, meaning that a person is keener to trust someone with common contacts or relationships, also known as a known entity.
- Trust: Trust as a social engineering principle is the effort that an attacker makes to develop a relationship with a victim.
- Distraction: People's tendency to get distracted can also be exploited by schemers. When someone's focus is diverted, he or she may ignore other things that are happening.
- Social proof: The use of social proof, meaning convincing the target that he acts in a way that is socially acceptable and has been done by others before, can ease his or her mind to make the decision to take an action they are not comfortable with.
- Urgency: A technique that applies pressure on the victim to act quickly without giving him or her the time to consider their actions.
- Greed-need: Greed and need are powerful motives in and of themselves. In this case, usually, an advance-fee fraud occurs, which involves the promise of wealth, gifts or employment in exchange for a payment.
- Phishing: A form of social engineering attack focused on stealing credentials or identity information from any potential target.

Social engineering is considered one of the strongest weapons in the armory of hackers and malicious code writers, as it is much easier to trick someone into giving his or her passwords for a system than to spend the effort to hack it [19,20]. A characteristic example is when Thomas R. Peltier and his team [21] were asked by a client if they could obtain employee access accounts and passwords. The company had an aggressive awareness campaign to remind employees of the need to keep their passwords from being compromised. Instead of installing a sniffer like the client supposed they would, the only thing that had to be made was a call to his employees. They called twelve employees, of whom nine answered their call. Posing as network administration staff, they asked for their account identification and password in order to troubleshoot a problem. Of the nine who answered the call, eight gave up the information. The ninth could not find the note on which he had written his password. The method Thomas R. Peltier and his team used in this example was a phishing attack. Phishing attack is the most preferable method of social engineers and as mentioned before, this kind of method aims to gain private and confidential information or make a victim act on behalf of the attacker. Phishing attacks can be classified into six categories according to [16,22]:

- Spear Phishing: These attacks require collecting information about the victim using available data online (OSINT).
- Whaling Phishing: A spear phishing attack with the difference that it targets high-profile targets.
- Vishing Phishing: This term refers to phone phishing, meaning phishing attempts performed via voice.

- Smishing: The term refers to phishing via SMS communication, usually used to load malware on mobile devices or steal credentials.
- Interactive Voice Response Phishing: These attacks are performed by using an interactive voice response system pretending to be a legitimate business or bank.
- Business Email Compromise Phishing: This term refers to scams that usually involve the compromise of legitimate business email accounts in order to deceive the victim to conduct unauthorized transfers of funds, giving confidential information [23,24], and in some cases, installing malware (such as keyloggers). An attacker often uses an address that may differ from that of the attacked organization by just one or two letters. For example, a scammer can replace a lowercase L ("l") with a capital I ("I"), making it extremely difficult to distinguish with the naked eye [25] (Figure 8).

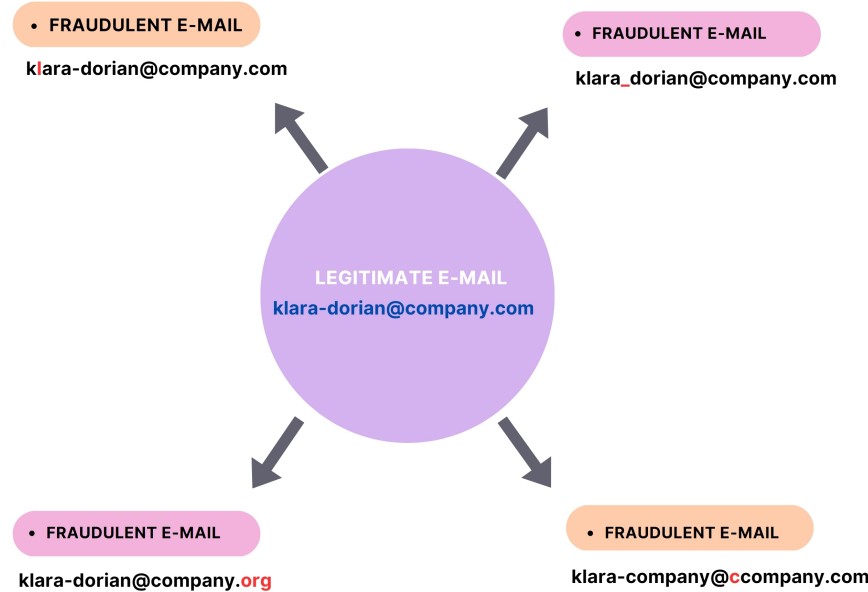

**Figure 8.** Cases of fraudulent email addresses. Notice that in the first case of the spoofed email, the only difference is the replacement of the lowercase L ("l") with the capital I ("I").

Social engineering is so successful and hard to defend against because it relies on the weakest link of all security systems, the human factor. As humans tend to trust other humans more easily, no hardware or software will ever be sufficient. Overall, a successful strategy against social engineering will not only require the appropriate technical systems, but also stiff policies and standards and a continuous education of personnel awareness.

## 4. Methodology Strategies Preventing BEC Attacks

### 4.1. Prevention—Non Technical Methods

Prevention is the first step for stopping BEC attacks from happening, and a key part in that is the investment in continuous employee training and defining policies. More thoroughly:

- Continuous employee training. Employees must be able to detect, report, and respond to a BEC attack. Sensitive departments of a company, such as the financial department, should be trained on a regular basis on social engineering techniques and BEC schemes. Employees must also be suspicious of hyperlinks, attachments, name misspellings, and last-minute directions on wire transfers or changed account details. The verification of the vendor's info should also be encouraged. All the above should take place, keeping in mind that social engineering and BEC schemes are evolving; that is why continuous and updated training is needed.
- Adopting new and innovative training techniques. Content relevance with technologies can vary in a company, as it does in society in general. Someone should not

expect a member of a company's accounting department to have the same familiarity with computer technologies as an IT member. For that reason, innovative training techniques like thematic and game-based analysis techniques are essential for a better understanding of the dangers lurking in online communication ([26]).

- Formation of blue and red teams is strongly advised, especially for big companies. Blue teams are responsible for evaluating and defending security environments, while red teams are responsible for attacking these environments in order to detect vulnerabilities.

- Defining policies: Establish internal regulations, policies and guidelines requiring special protections for sharing information and for wire transfers or other financial transactions. Make sure that email requests for transferring funds are forbidden and that financial transactions will require the presence of several persons or at least a vocal confirmation. As for communication via phone, setting requirements—like identity verification questions—should be enforced to prevent unwanted data leaks. Establishment of reports when incidents occur should also be encouraged.

- Fraud risk evaluation: Conducting fraud risk evaluation can detect and prevent vulnerabilities that scammers can exploit. An example of the above mentioned is the fraud risk management guide published by the Committee of Sponsoring Organizations for the Treadway Commission (COSO) partnered with the Association of Certified Fraud Examiners (ACFE) and also the ISO 22380:2018.

- Perform regular real world checkups: Given the fact that the employees had undergone a thorough education about social engineering and BEC schemes, and given that those policies had been established, regular real-world checkups should be performed by professional partners like pentesters and social engineers.

- Establishment of a social engineering department: A department with employees trained in social engineering and Open Source Investigations (OSINTs) is a must have, especially for big companies. Employees can utilize OSINT tools to conduct investigations on high-profile targets within their own company. These tools enable them to explore potential data breaches and leaks by leveraging online services such as "Have I Been Pwned," which are offered by Dehashed's online portal, a reputable organization headquartered on Market Street in San Francisco, CA, USA. All the above gathered information can be used by an attacker to d0x a target, so understanding the gaps and possible compromises in a company's profile is crucial to prevent/detect future BEC attempts [27].

- Compose safe lists containing email addresses of a company's employees.

### 4.2. Technical Methods

Prevention alone is a good start, but it will not suffice against BEC attackers and their evolving techniques. The second line of defense should technical methods, such as:

- Anti-spam and anti-malware software: Anti-spam software provides protection from spam and phishing attacks, while anti-malware software is able to detect malware. Both of the aforementioned attacks can be part of a BEC scheme.

- Time-of-click protection: This rewrites URLs in email messages and provides point-of-click protection using multiple reputation services [28].

- Executive tracking list: Uses details synched from an Active Directory to automatically detect users' real names within header and envelope address fields [28].

- Nearby domains: Compares the sender domain to legitimate domain names to identify nearby domains, meaning domains that vary by one or two characters [28].

- Directory harvest attack (DHA) prevention: Drops an email destined for invalid or fake email addresses [28].

- Multi-factor authentication (MFA): MFA provides a valid means for authentication that requires two or more verification factors to gain access to a resource [29].

- Email protocols: confirm that email protocols are updated and ban old email protocols (POP, IMAP, STMP) that can be used to override MFA.

- DKIM: DomainKey Mail (DKIM) is a cryptographic method for email integrity checks and authentication. Emails are signed by the private key of the sending domain, and the receiver domain verifies the signature of the email using the public key of the sender domain available through DNS.
- SPF: Sender Policy Framework (SPF) provides validation of an email's IP from a list of authorized IP addresses.
- DMARC authentication: Domain-based Message Authentication, Reporting and Conformance (DMARC) combines mechanisms that include DKIM and SPF, and it prevents attackers from spoofing the organization and domain [30]. Its main weakness is that DMARC processes only header information, and it is weak against impersonation.
- Encryption: Encryption can be used to prevent data breaches by requiring both the sender and receiver to have a pair of cryptographic keys.
- Verifiers: Using applications like an invoice verifier, which can scan a QR code on an invoice, provides a verification method that allows users to ensure the authenticity of the invoice attached in an email [31].
- Machine Learning (ML): machine learning technology, as part of the field of artificial intelligence, is composed by algorithms that can analyze email data to construct a model that can classify them and report any anomalies as red flags indicating BEC attempts [32].

While all the above counter measurements are required to successfully repel BEC attacks, the most promising and ever developing field is that of ML. ML algorithms can detect fraudulent email addresses, scan for red flags like the absence of DMARC, and learn, from email databases, the normal behavior of a user. There are four categories of machine learning algorithms ([33–35]), the selection of which can be determined by the needs of each company:

- Supervised learning: The operator provides the machine learning algorithm with a known dataset that includes desired inputs and outputs, and the algorithm must find a method to determine how to arrive at those inputs and outputs. The algorithm tries to model relationships between the target output and the input features and match the operator's results. This process continues until the algorithm achieves a high level of accuracy.
- Semi-supervised learning: Semi-supervised learning is similar to supervised learning in the way that it uses labeled data and is similar to unsupervised learning because it also uses unlabeled data. An operator can use a small amount of data to train a model to classify the rest of the unlabeled data in a dataset.
- Unsupervised learning: In this case, the algorithm studies data to identify patterns without the help of a human operator.
- Reinforcement learning: In reinforcement learning, an algorithm is provided with a set of actions, parameters and end values, but it must find the optimal action through its own experience. This method is based on rewarding desire behaviors and/or punishing undesired ones, meaning that the algorithm obtains the appropriate appraisal value of the environment state and revises its own strategy to adapt to the environment.

*4.3. Detailed Description of BEC Prevention Procedures in Greek Landscape*

In the past decade, Greece has suffered from the economic crisis, the outbreak of COVID-19, and more recently, from the consequences of the Russia–Ukraine war. The above mentioned factors are correlated to the increase in cybercrime incidents from the authors' point of view. As for BEC incidents in Greece, Figure 9 depicts the number of BEC incidents reported since 2020 in the Cyber Crime Division of the Hellenic Police ([36]). While there has been an increase in the number of cases for the year 2021 in comparison with 2020 due to the extensive lockdowns and the increase in people working from home, in 2022 and after the normalization of work, the number of cases fell almost to that of 2020.

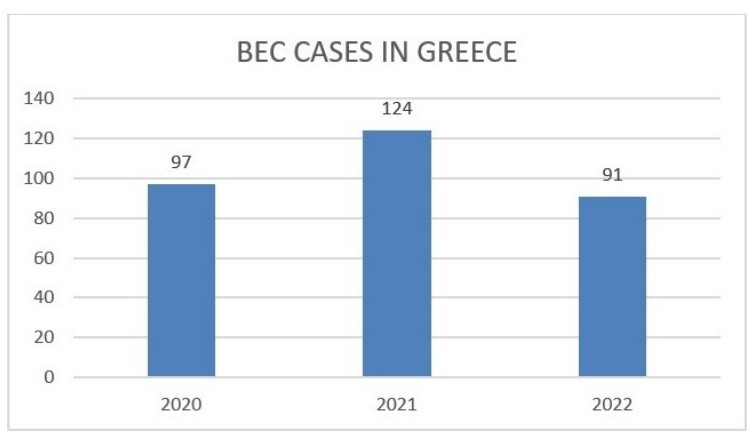

**Figure 9.** Number of BEC cases in Greece for the years 2020–2022.

Figure 10 depicts the log files and selected cloud companies for BEC frauds by the Hellenic Cybercrime Division for the years 2019–2022.

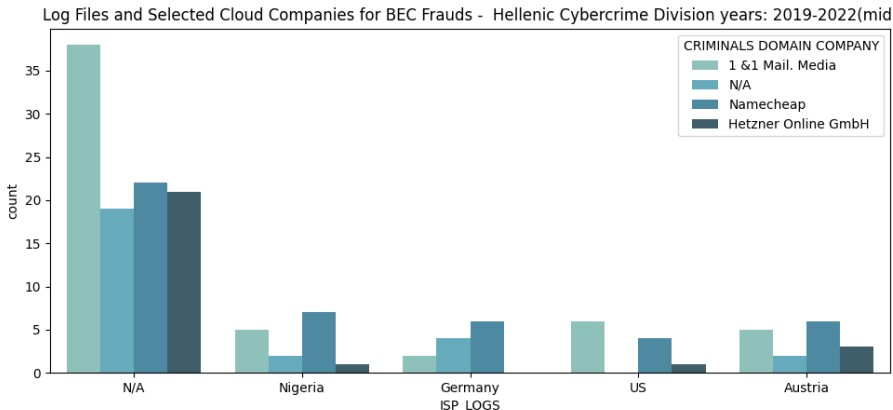

**Figure 10.** Log files and selected cloud companies for BEC frauds, for the years 2019–2022.

In the following sections, we report on some of the departments responsible for dealing with cybercrime in Greece and the Greek legislation related to cybercrime (Table 2), the actions needed after a BEC attack, and the compliance of Greece with NIS Directive 1 and 2.

**Table 2.** Greek departments responsible for cybersecurity and cybercrime and their legal tools for investigation of Business Email Compromise cases/crimes.

| Departments | Legal Tools |
|---|---|
| Greek Cyber Crime Division (GCCD) | Removal of the confidentiality of communications/lawful interception <br> Tracing the money route <br> Disclosing information about malicious users in the EU <br> Join Investigation Teams (JITs) in the EU <br> Disclosing information about malicious users outside the EU |
| Computer Emergency Response Team (CERT) | Removal of the confidentiality of communications/lawful interception <br> Disclosing information about malicious users in the EU <br> Disclosing information about malicious users outside the EU |
| Hellenic Computer Security Incident Response Team (CSIRT) | Disclosing information about malicious users in the EU <br> Disclosing information about malicious users outside the EU |
| National Cyber Security Authority (NCSA) | Disclosing information about malicious users in the EU <br> Disclosing information about malicious users outside the EU |

*4.4. Greek Departments Responsible for Cybersecurity and Cybercrime*

In Greece, the competent division for the prevention, response, and fight against cybercrimes is the Cybercrime Division of the Hellenic Police Headquarters. The Directorate for the prosecution of cybercrimes is based in Athens, and its territorial jurisdiction is nationwide. It functions as an independent central service, directly subordinate to the Chief of the Hellenic Police in accordance with the provisions of article 31 of Presidential Decree 178/2014. The Directorate for the Prosecution of cybercrimes is structured into the following departments:

4.4.1. Department of Administrative Support and Information Management

The main responsibilities of the Department of Administrative Support and Information Management are the following:

- Collection, study, analysis, evaluation and processing of information and data related to cases involving cybercrime. This department can also forward any processed data to the competent departments of the directorate for operational use.
- Investigating cases of suicide and cases of suicidal intent or disappearance via the internet.
- Provide assistance to relevant government agencies to prevent suicides reported online.

The Department has an operation center with a complaint line, as well as an electronic address (e-mail) for the service's communication to citizens, and a contact point for the network of the Council of Europe Convention on Cybercrime (Budapest Convention) and Directive 2013/40/EU for attacks against Information Systems (Article 6 of law. 4411/2016)

4.4.2. Department of Innovative Actions and Strategy

The main responsibilities of the Department of Innovative Actions and Strategy are the following:

- The preparation of information programs for citizens and institutions for internet and electronic crime issues through the implementation of various actions, such as conferences, workshops and videoconferences, as well as the organization of other innovative actions in the field of combating electronic crime;
- Defining strategic planning issues regarding cybercrime;
- The promotion and publication of the social work of the service through the creation and management of profiles on social networking sites (Twitter, Facebook, etc.) for the purposes of citizens' communication, information and awareness of issues of electronic threats and risks;
- The monitoring of developments in cybercrime issues, both domestically and internationally, the preparation of a relevant annual study with conclusions on the criminality of these offences in the country, and the submission of specific reasoned proposals to address them;
- The recording of actions and statistics regarding cybercrimes and their observance.

4.4.3. Department of Electronic and Telephone Communications Security and Software and Copyright Protection

The Department of Electronic and Telephone Communications Security and Software and Copyright Protection aims to prevent and repress crimes regarding the violation of the confidentiality of electronic communications and can cooperate with the Communications Confidentiality Authority as well as with other authorities, under the supervision of a competent prosecutor.

This department is also responsible for:

1. Handling cases of illegal infiltration in computer systems and theft, and the destruction or illicit trafficking of hardware, software, digital data and audiovisual works.
2. To assist other competent authorities investigating such cases, in accordance with applicable laws.
3. The provision of necessary technical assistance to other departments.

4. Conducting digital and internet research using modern technological equipment.
5. The analysis of digital data, files and other findings in criminal cases.

### 4.4.4. Department of Online Child Protection

The main responsibilities of the Department of Online Child Protection are the following:

- The detection and prosecution of crimes committed against minors through the use of the internet and other means of electronic or digital communication and storage.
- Investigating cases of online harassment (cyber bullying) in minors.

### 4.4.5. Department for the Prosecution of Online Financial Crimes

The main responsibilities of the Department for the Prosecution of Online Financial Crimes are the following:

- Combating economic crimes against the financial interests of the state and the national economy, in cooperation with the Directorate of financial police and other competent national, European and foreign services and authorities.
- The investigation of financial cybercrimes, if specialized technical or digital research is required.
- Assisting government agencies in investigating cases involving virtual/digital currencies.

### 4.4.6. Department of Special Affairs and Digital Investigation

The responsibilities of the Department of Special Affairs and Digital Investigation are as follows:

- The handling of serious and organized crime cases as well as conventional crimes committed via the internet, the investigation of which can only be carried out through specialized technical or digital research.
- The continuous investigation of the internet and other means of electronic communication and digital storage for the discovery, detection and prosecution of criminal offenses.

As we saw in Figure 6 Greece is listed in the top 20 countries by number of total victims as compared to the United States for 2021. Figure 11 shows the number of cases that Greek Cyber Crime Division handled in the period 2017–2022 and depicts the increased tension of cybercrime over the years.

Total number of cases handled by Cyber Crime Division

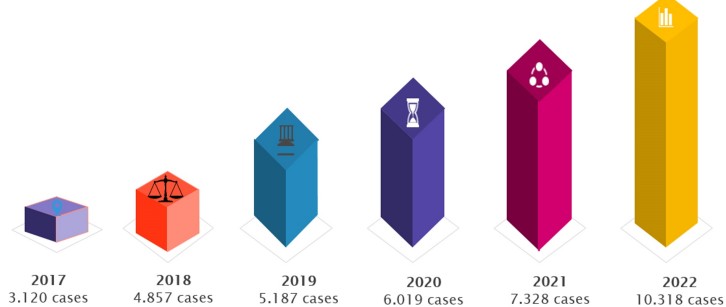

**Figure 11.** Number of cases that Greek Cyber Crime Division handled the period 2017–2022.

### 4.4.7. Computer Emergency Response Team (CERT)

The Computer Emergency Response Team (CERT) is part of the Hellenic National Intelligence Service (EYP). According to the 96/2020 (A'232) Presidential Decree, the responsibilities of the CERT are:

- The security of national communication and information technology systems and the evaluation–certification of classified devices and systems concerning communication and IT.

- The evaluation–certification of crypto systems and supporting the Greek Army and the public sector in issues of crypto security.
- The security of national electronic communication devices.
- Preventing, warning and dealing with cyberattacks against the Governance and the Ministries (except for the Defense Ministry).

### 4.4.8. Hellenic Computer Security Incident Response Team (CSIRT)

The Hellenic Computer Security Incident Response Team (CSIRT) is part of the Ministry of Defense. This department is responsible for the defense against cyber attacks in the military sector and aims to reduce the Nation's risk of systemic cybersecurity and communications challenges. CSIRT provides instructions and educational videos for the protection against BEC and social engineering attacks in general [37].

### 4.4.9. National Cyber Security Authority (NCSA)

The National Cyber Security Authority (NCSA) was established in the Ministry of Digital Governance, following a Presidential Decree. According to Greek National Law 4577/2018, the National Cyber Security Authority acts as the National Competent Authority for cybersecurity in Greece and collaborates with the relevant CSIRTs and all other national actors to achieve the national objectives of ensuring a high level of security for networks and information systems. NCSA, in close cooperation with the CSIRTs and other national authorities, assesses the technical and organizational measures implemented by the Operators of Essential Services (OES) and other entities, handles all critical incidents, issues binding instructions, and imposes corrective actions and penalties when necessary. As the national representative in international and European organizations and working groups for cybersecurity, the NCSA opens channels of communication with strategic stakeholders and promotes all necessary steps towards a secure cyberspace [38].

### 4.5. Legal Framework and Investigative Tools for Business Email Compromise Cases/Crimes

4.5.1. National Legal and Regulatory Framework for Preventing, Combating and Prosecuting Business Email Compromise Cases/Crimes

Cases of online fraud using the Business Email Compromise method are criminalized and punished by Greek law according to Article 386 A of the Criminal Code, entitled "Computer Fraud", which states that anyone who, in order to accede to himself or another illegal property benefit, damages foreign property by affecting the result of a computer data processing process, is punished with imprisonment, while if the damage caused exceeds a total amount of EUR 120,000, imprisonment is imposed.

Furthermore, in order to commit the above illegal acts, illegal access to an information system is carried out, which is criminalized by Greek legislation according to Article 370 B of the Criminal Code, entitled "illegal access to an Information System or data", which stipulates that anyone who, in violation of a protection measure and without right, gains access to part of or an entire information system or private dataset is punishable by imprisonment up to two years or a fine, while if there are confidential company data (e.g., within the email that was violated), those acts are punishable by imprisonment up to three years or a fine. In addition, in these cases of online fraud, personal data files (e.g., an e-mail account) are compromised in violation of the current law 4624/2019 "on personal data". Additionally, the perpetrators of the aforementioned types of fraud are also considered to be accountable for the crime of money laundering because they use the financial/banking sector for transferring revenues derived from illegal/criminal activities with the ultimate aim of giving legitimacy to these revenues in violation of the current legislation for the prevention and suppression of money laundering 4557/2018 and N. 4734/2020. In the above crimes (article 386 A of the Criminal Code), it should be noted that the crime is eliminated (in accordance with article 406 A PK) if the perpetrator of his own will, and before being examined in any way for his action by the authorities, completely satisfies the injured party. This is considered a major throwback in fighting cybercrime because it

advocates actors taking the risk of committing this type of criminal activity due to the prior knowledge that even if the actor is detected and arrested, satisfying the injured person will result in the elimination of the crime.

### 4.5.2. Legal Tools for Investigation of Business Email Compromise Cases/Crimes and International Cooperation

Removal of the Confidentiality of Communications/Lawful Interception

For the effective investigation of BEC cases/crimes, the first step is to remove the confidentiality of communications and obtain all the necessary data from the domestic internet service providers. The removal of the confidentiality of communications, according to relatively recent legislation (Law 5002/2022 "procedure for the removal of the confidentiality of communications, cybersecurity and protection of personal data of citizens"), is permissible for the verification of various offenses, including computer fraud (article 386 A of the Criminal Code). The confidentiality of communications does not cover communication via the Internet (Internet) and external contact details (names and other subscriber data, telephone numbers, time and place of call, duration of conversation, etc.), and for that reason, the prosecuting, investigating and pre-investigation authorities are entitled to request, from the providers of internet and communication services, the electronic traces of a criminal act; the chronology and the details of the person to whom the electronic trace corresponds; and from the other providers of communication services, the "external data" of communication, while the provider is obliged to deliver them without the need for prior permission of an authority, in accordance with current opinions of the prosecutor's Office of the Supreme Court (opinions of the prosecutor's office of the Supreme Court 9/2009, 12/2009 and 9/2011).

Tracing the Money Route

Law Enforcement Agencies (LEAs) apply a basic investigative practice and technique in cases of crime investigation, especially those with financial impact and property damage, in accordance with the American doctrine that dictates: "follow the money". This can be achieved either by removing bank/financial secrecy or by investigating financial data and transactions. The procedure is facilitated and significantly accelerated in practice, without the intervention of a judicial council, by applying the provisions of N. 4557/2018, that is, only with a simple written express order from the prosecutor conducting the investigation into such crimes, without the interference of a court or a judicial council.

Disclosing Information about Malicious Users in the EU

Authorities' access to electronic evidence is a complex and time-consuming issue, particularly due to not only the different legislations in force in the member states of the European Union, but also due to the place where the data are stored. So, in the case of investigating such cases that electronic evidence arises (e-mail addresses, internet protocol addresses (IPS), etc.) in a country of the European Union, data can be moved from one country of the European Union to another if the procedure is followed in accordance with directive 2014/41/EU of the European Investigation Order (EIO), which was incorporated into the Greek legal order by law 4489/2017 and is the most comprehensive regulation to date in the field of assistance between EU member states providing effective and quick access to electronic evidence. Time of investigation will be accelerated even more with the advent of the common European framework for access to electronic evidence, in particular the E-evidence Regulation, which will enable police and judicial authorities to obtain more quickly and easily the electronic evidence they need.

Join Investigation Teams (JITs) in EU

The Joint Investigation Teams (JITs) serve as an alternative tool for EU investigations. These teams are highly advanced and are employed for international cooperation in criminal matters. They involve a legal agreement between competent authorities from two or

more states, enabling them to conduct criminal investigations jointly. The establishment of JITs between member states of the European Union is governed by Article 13 of the 2000 EU Convention on Mutual Assistance in Criminal Matters (2000 EU MLA Convention) and the 2002 Council framework decision on JITs. Additionally, the Joint Cybercrime Action Taskforce (J-CAT) model is another option available. The J-CAT operates under the jurisdiction of the European Cybercrime Centre (EC3) at Europol [39].

Disclosing Information about Malicious Users Outside EU

When electronic evidence is located in a third country outside the EU, then in accordance with Article 458 of the Greek code of Criminal Procedure, a request for legal assistance from the Greek judicial authorities to the foreign authorities for the seizure of evidence, the operation of an autopsy and expert examination, and other investigative acts provided for in the CPC must be submitted by the competent prosecutor to the Ministry of justice, transparency and Human Rights, which ensures its execution through the Ministry of Foreign Affairs, based on the provisions of international treaties and customs. Mutual Legal Assistance Treaties (MLATs), International Letters of Request (ILORs), or Letters Rogatory, which are commonly known as "judicial assistance", are the official means through which states seek and offer assistance in acquiring evidence located in one state to aid criminal investigations or legal proceedings in another state. Mutual legal assistance primarily focuses on gathering evidence and does not encompass intelligence or other types of information gathering.

### 4.6. Procedures after a BEC Attack

The general steps from the attacker's point of view is to gain access to a victim's credentials, impersonate a trusted entity, send fraudulent email, and extract money or sensitive information, as shown in Figure 12.

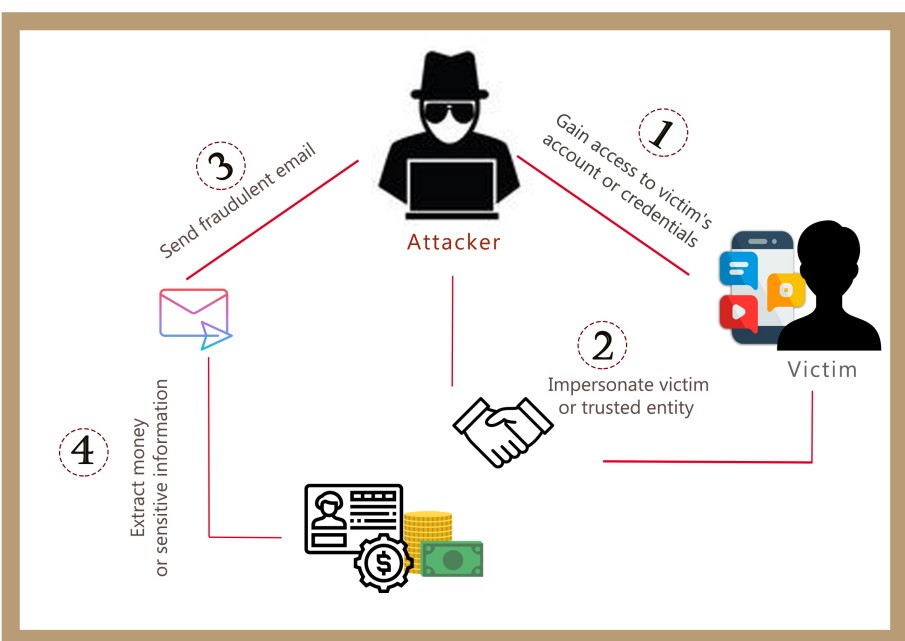

**Figure 12.** General of a BEC attack from the attackers point of view.

After a cyberattack in general, and in this case after a BEC attack, it is crucial for the victim to perform a number of procedures in order to inform and facilitate the relevant authorities (Figure 13). These procedures include [40]:

- Contacting law enforcement (in the case of Greece, the Unit of Special Cases and Internet Economic Crimes Prosecution) and other competent authorities as soon as possible.

- Immediately informing the banking institute via phone and then through a Transaction Dispute Form.
- Providing the authorities dealing with the case with evidence like:
  - Email addresses, email headers, and associated Internet Protocol (IP) addresses with their respective timestamps.
  - A description and timing of suspicious email communications.
  - Phone numbers.
  - Wire transfer details:
    * Dates and amounts of suspicious transactions.
    * The sender's identifying information, account number and financial institution.
    * The beneficiary's identifying information, account number and financial institution.
    * The correspondent and intermediary financial institution's information, if applicable.

From this point and onward, the Law Enforcement Agencies (LEAs) and the investigators will alert the Beneficiary Financial Institution and the Financial Intelligence Unit (FIU), and they will try to analyze the BEC attack by:

- Obtaining a copy of the emails and performing the following steps:
  - Initiating the investigation process by requesting information from the victim's email provider (if applicable). This entails:
    * IMAP (Internet Message Access Protocol)—The emails must remain on the server. A copy of the emails should be obtained from the email provider via the legal process or from the victim's server.
    * POP (Post Office Protocol)—Emails are downloaded from the email provider to the victim's computer. Emails are not located on an external server and must be obtained from the victim's computer.
    * An investigation for spear phishing attacks from what appears to be legitimate vendors (slight changes in email address) and for additional communications with the misspelled domain.
    * An investigation for indications of out-of-band communication changes (e.g., phone number).
    * An investigation for possible email redirect rule.
    * Running a PowerShell command across the network looking for forwarding rules because more than one computer may be affected.
  - Forensic Analysis of the computer:
    * Malware may have been used to harvest email credentials (allowing access to the email account). The use of malware may or may not be contemporaneous with the BEC. Often, access to the email account occurs weeks/months prior to the BEC.
    * An analysis of the victim's computer/system, even though it does not generally lead to direct information of the transfer of funds (usually provides information on how the email credentials were compromised).

The above procedure is common among the EU members. In later stages of the investigation and as mentioned before, especially if the attack or the wire transfers originated from foreign countries, a Joint Investigation Team can be formed. Knowing how to treat the evidence of a BEC scheme is crucial for the successful investigation of the crime. Companies, as mentioned previously in the section of non-technical methods of protection against BEC attacks, should have well trained and educated employees (preferably a blue team with forensic responsibilities) who can detect such attacks and gather as much evidence possible before LEAs take over the investigation [41].

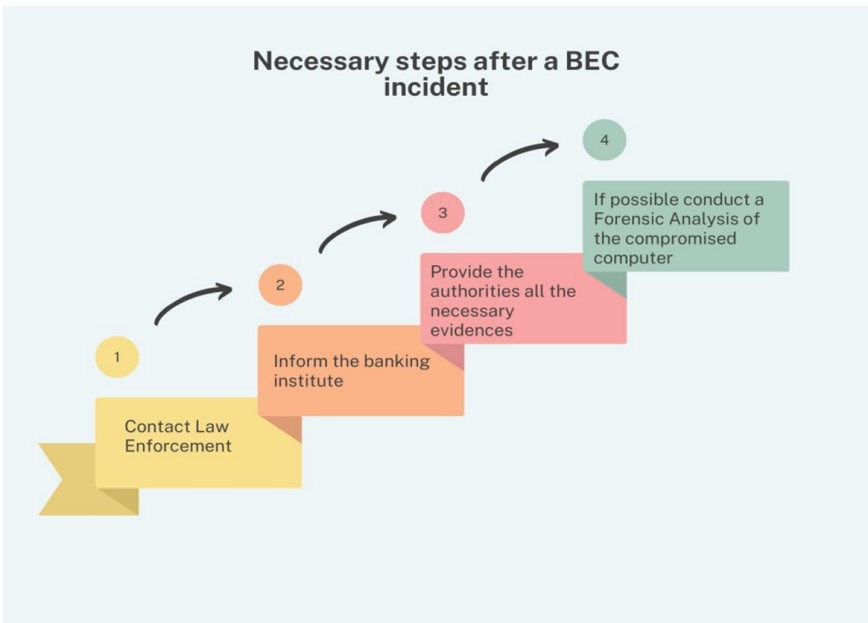

**Figure 13.** Necessary steps after a BEC incident.

*4.7. NIS Directive and Greek Compliance*

The directive for the security of networks and information systems (NIS Directive) provides legal measures to increase the level of cybersecurity in the European Union. The NIS1 Directive (EU 2016/1148) was the first piece of EU-wide cybersecurity legislation and was adopted in 2016. Every member state in the EU has adopted its national legislation to follow the directive [42]. The main obligations that derived from the NIS1 directive for Greece and every member state were [43,44]:

- Each member state is obliged to participate with representatives in the Cooperation Group and the CSIRT network.
- Each member state is obliged to establish a national strategy for the security of networks and information systems in order to achieve a high level of security that keeps up with the current technological advances and challenges.
- Each member state must designate one or more national competent authorities for the security of networks and information systems.
- Each member state is obliged to appoint one or more CSIRTs responsible for managing risks and handling incidents.
- Member States are obligated to establish regulations regarding penalties for violations of national provisions enacted under this directive and must undertake all necessary actions to ensure their enforcement. The prescribed penalties should be impactful and fair and should serve as a deterrent.
- Member states are required to identify, by November 2018, the entities operating essential services.
- Starting in August 2018, and subsequently on an annual basis, it is mandatory to submit a summary report of the notifications received by the Cooperation Group.

According to Maglaras et al. [45], in order to comply with the NIS directive, Greece has taken the following steps:

- The National Cyber Security Authority (NCSA) was established by the Presidential Decree of 82/2017, along with the Single Point of Contact under the Ministry of Digital Policy, Telecommunications and Media (now Ministry of Digital Governance).
- The NCSA represents Greece in the Cooperation Group and the National CERT in the CSIRT network of the EU.

Furthermore, from 2018 and onwards:

- A National Cyber Security Strategy was published in 2018 [45].
- Greek Parliament enacted Law 4577/2018, establishing the national cybersecurity plan and cybersecurity authority in the Ministry of Digital Policy, Telecommunications and Media (now Ministry of Digital Governance) and also setting a range of network and information security requirements for the Operators of Essential Services (OESs) and Digital Service Providers (DSPs).
- Law 4577/2018 also defined the penalties in case of non-compliance.
- The first Annual Report NIS Directive Incidents 2019 was published in December 2020 [46].

In December 2021, the European Commission adopted a proposal for a revised Directive on Security and Information Systems—the NIS2 Directive [47]. NIS2 aims to address the deficiencies of the previous NIS1 Directive. The key elements of NIS2 are:

- Adding new sectors based on their criticality for the economy and society and introducing a clear size cap—meaning that all medium and large companies in selected sectors will be included.
- The removal of the differentiation between operators of essential services and digital service providers involves classifying entities based on their significance and dividing them into categories of "essential" and "important." This classification would result in subjecting them to distinct supervisory regimes.
- Enhanced security requirements for companies by implementing a risk management approach that mandates the application of a minimum list of fundamental security elements. This introduces more specific provisions regarding the incident reporting process, including the content of the reports and the required timelines.
- Emphasis of the importance of secure supply chains and supplier relationships by mandating that individual companies address cybersecurity risks within their supply chains and supplier relationships. At the European level, this enhances supply chain cybersecurity measures for important information and communication technologies. In collaboration with the Commission and ENISA, Member States will conduct synchronized risk assessments of critical supply chains.
- Implementation of stricter oversight measures for national authorities, the imposing of more rigorous enforcement requirements, and the aim to standardize sanction regimes among member states.
- The strengthening of the influence of the Cooperation Group in shaping strategic policy decisions concerning emerging technologies. It promotes greater information sharing and cooperation among authorities of member states. Additionally, it improves operational collaboration, particularly in the realm of cyber crisis management.
- The establishment of a fundamental framework to facilitate coordinated vulnerability disclosure for recently identified vulnerabilities throughout the EU. This also establishes an EU registry for this task, which will be operated by the European Union Agency for Cybersecurity (ENISA) [48].

In December 2022, the European Commission adopted a proposal for a revised Directive on Security and Information Systems—the 2022/2555 Directive [49]. The new Directive sets higher standards for cybersecurity for enterprises and the public sector, and it also makes accountable the executives of companies/enterprises in case of noncompliance. The key elements of the 2022/2555 Directive are:

- The requirement that member states must adopt national cybersecurity strategies and designate or establish competent authorities, cyber crisis management authorities, single points of contact for cybersecurity (single points of contact), and computer security incident response teams (CSIRTs).
- Requirements for entities in terms of managing cybersecurity risks for the following sectors: energy, transportation, banking, financial market infrastructures, health, drinking water, waste water, digital infrastructures, public administration, space, postal and courier services, waste management, manufacture production and distribution of

chemicals, manufacture production and distribution of food, manufacturing, digital providers, and research.
- Regulations and responsibilities regarding the sharing of cybersecurity information.
- Obligations for member states in terms of supervision and enforcement.
- The establishment of the European Cyber Crisis Liaison Organization Network (EU-CyCLONe), which aims to facilitate the coordinated handling of significant cybersecurity incidents and crises. This also ensures the consistent exchange of information between member states and various union institutions, bodies, offices, and agencies.
- Specifies the overall criteria for imposing administrative fines on entities classified as essential and important.

Greece's compliance with the Network and Information Systems (NIS) Directives can help to establish a strong cybersecurity framework that protects against a range of cyber threats, including Business Email Compromise (BEC) attacks. By implementing the NIS Directives, Greece can take proactive steps to reduce the risk of BEC attacks and enhance the security of its critical infrastructure. For example, one of the key measures taken when complying with the NIS directives is the implementation of multi-factor authentication (MFA) for email communication systems. MFA requires users to provide more than one form of identification to access their email account, such as a password and a verification code sent to their mobile phone. This additional layer of security can help to prevent unauthorized access to email accounts and reduce the risk of BEC attacks. Another important measure is the encryption of sensitive information transmitted via email. By encrypting emails containing sensitive information, Greece can ensure that the information remains confidential and is only accessible to authorized individuals. This can help to prevent cybercriminals from intercepting emails containing sensitive information and using them to carry out BEC attacks. In addition to these technical measures, compliance with the NIS directives also involves providing security awareness training to employees. This training can help to educate employees on the risks of BEC attacks and other forms of cybercrime, and it can provide them with the knowledge and skills necessary to identify and report suspicious activity. By promoting a culture of cybersecurity awareness, Greece can further reduce the risk of BEC attacks and enhance the overall security of its critical infrastructure. The NIS2 directive and the 2022/2555 directive have established higher cybersecurity standards for organizations and the private sector. One effective way to comply with these standards is by implementing ISO 27001/2022 (Appendix A, Table A1). ISO 27001 is a valuable tool for any business or organization in their fight against Business Email Compromise. This is because the standard helps organizations to:

- Identify information and cybersecurity risks.
- Establish technical control measures to manage or mitigate the risks of Business Email Compromise, such as employing strong authentication practices, setting up multifactor authentication (MFA) to add an extra layer of security, restricting editing and signing rights, organizing security training for employees, setting up email filters and blocks, verifying all payment requests, monitoring email accounts, using encryption and secure communication channels, and keeping software and security measures up to date.
- Implement procedures for the prompt detection of business email compromise attacks.
- Recover business operations following an incident like Business Email Compromise.

In summary, compliance with the NIS directives can help Greece to establish a comprehensive cybersecurity framework that protects against a range of cyber threats, including BEC attacks. Measures such as the implementation of MFA, the encryption of sensitive information, and security awareness training can all contribute to a more secure email communication system and reduce the risk of cybercrime [50,51].

## 5. Results

We have demonstrated that Business Email Compromise (BEC) attacks pose a persistent and evolving threat to individuals, enterprises, and critical infrastructures. The losses incurred by these attacks are consistently increasing, paralleled by the complexity of the schemes employed. Successful execution of a BEC scheme requires a combination of technical skills and social engineering techniques. However, recent years have seen a decrease in required technical expertise, thanks to the readily available information on the dark web and the Internet in general. Nevertheless, sophisticated BEC attacks still incorporate social engineering methods alongside technical skills, as discussed in previous sections. To counter BEC attacks, we have outlined a range of non-technical methods, such as employee education, policy and regulation development, and creating safe lists. Additionally, we have proposed various technical measures, including anti-spam and anti-malware software, multi-factor authentication, email protocols, DKIM, and machine learning algorithms. The combination of these non-technical and technical approaches forms a robust defense against BEC attacks, making it significantly more challenging for attackers to succeed. Furthermore, we have provided insights into the procedures that follow a BEC attack and suggested steps to facilitate the work of authorities. These guidelines are intended to aid individuals who may be unfamiliar with such schemes, offering valuable guidance on necessary actions and reporting mechanisms. In addition to the aforementioned measures, countering cybercrime and BEC attacks necessitates the implementation of a comprehensive legal framework within each country. Taking Europe as an example, we have chosen Greece to illustrate its efforts in addressing cybercrime through various divisions and through harmonization with European legislation. Our research explores the legal and investigative tools available to combat cybercrime, including BEC schemes, in a broader context. The alignment of Greece with other EU countries in terms of legal frameworks can be attributed to European directives that have shaped legislation in this field. We have discussed these legal measures, such as NIS 1, NIS 2, Directive 2022/2555, and ISO 27001/2022, which apply to all EU member states. By highlighting these measures, we have underscored the significance of a universal legal framework in effectively tackling the escalating activity of cybercrime.

## 6. Future Work

Business Email Compromise (BEC) attacks can cause significant damage to businesses and organizations, and they require effective countermeasures. To address this issue, our team is currently developing a technical solution for large companies that will provide a unique signature of the sender's identity by including a QR code in the email. The QR code will contain encrypted information such as the MAC address of the sender's computer, username, IP address, and timestamp. When a receiver receives an email with sensitive information, our program will decrypt the information contained in the QR code and compare it with a pre-approved list of MAC and IP addresses and usernames. The program will also ensure that the timestamp is unique for that specific information. This approach has the potential to protect companies against BEC attacks, and is expected to be highly effective in practice. It is crucial to develop effective countermeasures against BEC attacks, and our technical solution offers a promising solution for large companies. As we continue to develop and test this solution, we are optimistic that it will prove to be a valuable tool in protecting businesses and organizations from the devastating effects of BEC attacks.

## 7. Discussion and Conclusions

Business Email Compromise (BEC) attacks are a major growing and developing threat for organizations and individuals, and they lead to significant economic losses and, in many cases, the paralysis of critical infrastructures. As we showed earlier, these sophisticated cyberattacks exploit vulnerabilities within email systems, using highly targeted and socially engineered tactics to deceive victims. The financial impact of BEC attacks cannot be underestimated. Organizations and individuals have suffered substantial monetary losses, ranging from thousands to millions of dollars per incident. Attackers exploit weaknesses

in financial processes, executing unauthorized wire transfers, fraudulent invoice payments, and illicit changes to banking information. These illicit actions cause severe economic harm to victims, often resulting in severe financial setbacks and operational disruptions. Moreover, BEC attacks inflict substantial reputational damage on organizations. By impersonating key executives or trusted entities, attackers erode trust and credibility. This loss of trust can lead to customer attrition, damaged relationships, and a tarnished market image. Rebuilding a reputation after a successful BEC attack can be a challenging process. In addition to financial and reputational repercussions, BEC attacks can wreak havoc on critical infrastructures. Through unauthorized access to email accounts, attackers can infiltrate internal systems, compromise sensitive data, and potentially disrupt essential operations. This disruption, whether through system downtime, data breaches, or operational failures, can have far-reaching consequences, particularly in sectors where uninterrupted operations are crucial, such as healthcare, transportation, energy, and finance. Mitigating the threat of BEC attacks poses an ongoing challenge for organizations and individuals alike. Attackers continually refine their tactics, employing advanced phishing techniques, social engineering, and compromised or spoofed email accounts to deceive victims. Exploiting human vulnerabilities and leveraging trust, these attackers successfully circumvent traditional security measures. As stated before, to effectively counter the threat of BEC attacks, organizations and individuals must prioritize cybersecurity awareness and adopt comprehensive defense strategies. This includes non-technical methods like educating employees about recognizing and reporting suspicious emails, conducting risk evaluation reports and harmonization with ISO 27001:2022, and technical methods like implementing robust access controls, regularly updating and patching systems, deploying email authentication mechanisms, using anti-spam and anti-malware software, and leveraging advanced threat detection technologies like machine learning algorithms. In our study, we aimed to shed light on the issue of Business Email Compromise (BEC) attacks. We delved into the intricacies of the BEC scheme and the social engineering techniques employed, aiming to enhance the audience's comprehension of this scheme. Additionally, we sought to offer a practical guide outlining effective measures like the above-mentioned measures that individuals and companies can adopt to safeguard themselves against BEC attacks. By exploring the tactics used in social engineering, we aimed to offer valuable insights into the psychological manipulation techniques employed by attackers. This deeper understanding can empower individuals and organizations to recognize and respond effectively to BEC attacks, mitigating the risk of falling victim to such deceptive schemes. Recognizing the importance of proactive defense, we dedicated a significant portion of our study to proposing actionable measures for protection. On top of that, in our research, we conducted a comprehensive analysis of the case of Greece as a representative example within the context of our study. Greece's inclusion in the top 20 countries with a significant number of BEC attack victims, while being a member of the European Union, highlights its relevance from the authors' perspective. By examining the departments and authorities involved in addressing cybercrime, we aimed to provide the general framework in Greece and other EU countries, demonstrating a cooperative approach across Europe to combat this issue. This collaboration includes the exchange of information and the establishment of Joined Investigation Teams, which are relatively recent developments aimed at enhancing cybercrime response capabilities. Furthermore, our paper explores the legal and investigative tools available for combating cybercrime in general, including BEC schemes. The alignment of Greece with other EU countries in terms of legal frameworks can be attributed to the European Directives that have shaped legislation in this domain. We discussed these legal measures, such as NIS 1, NIS 2, Directive 2022/2555, and ISO 27001/2022, which apply to all EU member states. By highlighting these measures, we emphasize the importance of a universal legal framework in addressing the rising losses caused by malicious cyber activities, especially BEC attacks. Additionally, we outlined the procedures that follow a BEC attack and proposed steps to facilitate the work of authorities. These guidelines aim to assist individuals who may be unfamiliar with this type of scheme, providing valuable

insights into the necessary actions and reporting mechanisms. While the departments responsible for addressing cybercrime exhibit similarities across EU member states, legislation in EU member states regarding cybercrime exhibited a lot of differences, and in many cases—as in the case of Greece—it was outdated. The need for a comprehensive and uniform legal framework became evident as losses due to cyber activities, particularly BEC attacks, escalated. Therefore, we emphasize the importance of the legal measures adopted by the European Union, which have proven essential in safeguarding individuals, enterprises, and critical infrastructures. Measures such as NIS 1, NIS 2, Directive 2022/2555, and ISO 27001/2022 have had a significant impact on protecting the interests of all member states, including Greece, and they serve as a benchmark for effective cybersecurity practices. Overall, our study endeavors to provide a comprehensive and analytical examination of the BEC threat landscape. By incorporating statistics, dissecting the BEC scheme and social engineering techniques, and offering practical protective measures, we aim to equip the readers with a well-rounded understanding of the problem and empower them to take proactive steps to safeguard themselves against BEC attacks. Additionally, by examining the case of the Greek landscape, we provide a comprehensive analysis of the legal, investigative, and cooperative initiatives aimed at addressing BEC attacks and cybercrime. The insights gained from this research contribute to a better understanding of the measures necessary to protect individuals, organizations, and critical infrastructures across all EU member states.

**Author Contributions:** Conceptualization, A.P. and G.L.; writing—original draft preparation, A.P. and G.L.; writing—review and editing, A.P., G.L., V.L. and E.G.; supervision V.L. and E.G. All authors have read and agreed to the published version of the manuscript.

**Funding:** This research received no external funding.

**Data Availability Statement:** Not applicable.

**Acknowledgments:** We acknowledge support of this work from the project "Immersive Virtual, Augmented and Mixed Reality Center of Epirus" (MIS 5047221) which is implemented under the Action "Reinforcement of the Research and Innovation Infrastructure", funded by the Operational Programme "Competitiveness, Entrepreneurship and Innovation" (NSRF 2014–2020) and co-financed by Greece and the European Union (European Regional Development Fund).

**Conflicts of Interest:** The authors declare no conflict of interest.

## Appendix A

**Table A1.** NIS 2 Cybersecurity risk-management measures mapping to ISO 27001:2022.

| NIS 2 | ISO 27001:2022 |
|---|---|
| Article 21.2 (a) Policies on risk analysis and information system security | 5.2 Policy<br>6.1.2 Information security risk assessment<br>6.1.3 Information security risk treatment<br>8.2 Information security risk assessment<br>8.3 Information security risk treatment<br>*Annex A. Information security controls reference:*<br>5.1 Policies for information security |
| Article 21.2 (b) Incident handling | *Annex A. Information security controls reference:*<br>5.24 Information security incident management planning and preparation<br>5.25 Assessment and decision on information security events<br>5.26 Response to information security incidents<br>5.27 Learning from information security incidents<br>5.28 Collection of evidence<br><br>6.8 Information security event reporting |

**Table A1.** *Cont.*

| NIS 2 | ISO 27001:2022 |
|---|---|
| Article 21.2 (c) Business continuity, such as backup management and disaster recovery, and crisis management | *Annex A. Information security controls reference:*<br>5.29 Information security during disruption<br>5.30 ICT readiness for business continuity<br>8.13 Information backup<br>8.14 Redundancy of information processing facilities |
| Article 21.2 (d) Supply chain security, including security related aspects concerning the relationships between each entity and its direct suppliers or service providers | *Annex A. Information security controls reference:*<br>5.19 Information security in supplier relationships<br>5.20 Addressing information security within supplier agreements<br>5.21 Managing information security in the ICT supply chain<br>5.22 Monitoring, review and change management of supplier services<br>5.23 Information security for use of cloud services |
| Article 21.2 (e) Security in network and information systems acquisition, development and maintenance, including vulnerability handling and disclosure | *Annex A. Information security controls reference:*<br>5.37 Documented operating procedures<br>8.8 Management of technical vulnerabilities<br>8.9 Configuration management<br>8.20 Network security<br>8.21 Security of network services |
| Article 21.2 (f) Policies and procedures to assess the effectiveness of cybersecurity risk management measures | 9.1 Monitoring, measurement, analysis and evaluation<br>9.2 Internal audit<br>9.3 Management review<br>*Annex A. Information security controls reference:*<br>5.35 Independent review of information security |

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
