# Peer review of "Business Email Compromise (BEC) Attacks: Threats, Vulnerabilities and Countermeasures—A Perspective on the Greek Landscape"

_jcp, doi:10.3390/jcp3030029_

Round 1

Reviewer 1 Report

The article initially provides background information on BEC and later discusses BEC cases in Greece.

The paper is well-written and organized for easy comprehension.  

However, there is a lack of evidence and citations to support the authors' claims. For instance, the authors mention the potential use of ML for detecting BEC attacks but fail to provide any sources or citations.  

The same issue arises in Section 5, where information and citations about the Greek cases are absent.  

Finally, to enhance audience understanding, it would be beneficial to include a figure illustrating the common attack vectors for BEC from the attackers' perspective, similar to Figure 11. This would provide a more comprehensive understanding of the BEC attack campaign.

Author Response

We would like to express our heartfelt gratitude for taking the time to review and provide feedback on our work. We truly appreciate your effort in helping me enhance the quality of our output. We have carefully considered each of your comments and suggestions, and we are pleased to present our replies to your comments below. 

Comment #1.1 The article initially provides background information on BEC and later discusses BEC cases in Greece. The paper is well-written and organized for easy comprehension.  However, there is a lack of evidence and citations to support the authors' claims. For instance, the authors mention the potential use of ML for detecting BEC attacks but fail to provide any sources or citations. 

Answer #1.1: We appreciate the above-mentioned observation and thank you for highlighting the fact that there is a lack of references regarding machine learning. We added the following references in page 13 which we thought to be relevant. The first is a very promising work for defending against BEC and the second a review.

References

  • CIDON, Asaf, et al. "High Precision Detection of Business Email Compromise." Journal of Cybersecurity and Privacy, 2019
  • ATLAM, Hany F.; OLUWATIMILEHIN, Olayonu. "Business Email Compromise Phishing Detection Based on Machine Learning: A Systematic Literature Review." Electronics, vol. 12, no. 1, 2022

 Attached is our response, along with the revised version of our article, highlighting all the improved sections in yellow.

Comment # 1.2 The same issue arises in Section 5, where information and citations about the Greek cases are absent.

Answer #1.2: Thank you for pointing the necessity for citation about the Greek cases. We now cite the Crime Division of Hellenic Police.   

Comment #1.3 Finally, to enhance audience understanding, it would be beneficial to include a figure illustrating the common attack vectors for BEC from the attackers' perspective, similar to Figure 11. This would provide a more comprehensive understanding of the BEC attack campaign

Answer #1.3: Thank you for the observation, we constructed the a figure that provides the reader with the perspective of an attacker and the general steps of a BEC scheme and we included it in the revised version of the paper as Figure 11.  

Reviewer 2 Report

  1. Authors are encouraged to use structured abstracts. (Context, Methodology, Objective, Results, and Conclusion).

  2. The paper's readability needs to be improved.

  3. Rubia Fatima and her research group have worked on countering Social Engineering attacks using games. A section mentioning strategies misses that perspective. The authors are encouraged to read her work and cite it if relevant. (P1: How persuasive is a phishing email? A phishing game for phishing awareness) (P2: Strategies for counteracting social engineering attacks) etc.

  4. At the end of the introduction section, the authors are advised to add a flow diagram that guides the reader on how different sections are linked. This will help the reader understand the paper flow for better readability.

  5. Figure 2 is taken from the source mentioned in the caption. Rather than using the same image, it would be better to refer to the same paper in the text.

  6. It seems that author's have written a report instead of paper. Authors are advised to use a template which is more readable to the reader. such as : Section 1: Introduction, Section 2: Literature Review, Section 3: Experiment Design, Section 4 : Results and Analysis, Section 5: Discussion and Conclusion. etc

Author Response

We would like to express our heartfelt gratitude for taking the time to review and provide feedback on our work. We truly appreciate your effort in helping me enhance the quality of our output. We have carefully considered each of your comments and suggestions, and we are pleased to present our replies to your comments below. 

Attached is our response, along with the revised version of our article, highlighting all the improved sections in yellow.

Comment #2.1 Authors are encouraged to use structured abstracts. (Context, Methodology, Objective, Results, and Conclusion).

Answer #2.1: Thank you for your comment. Having in mind that others reviewers also made that observation, we reformed the structure in the revised version and we hope that it will fit those standards.

Comment #2.2 The paper's readability needs to be improved.

Answer #2.2: We appreciate your comment. We reformed the structure of the paper, added some informational figures and added the following sections:

  • Contribution
  • Research Methodology
  • Results

We sincerely hope that those changes improved the readability of the paper.   

Comment #2.3: Rubia Fatima and her research group have worked on countering Social Engineering attacks using games. A section mentioning strategies misses that perspective. The authors are encouraged to read her work and cite it if relevant. (P1: How persuasive is a phishing email? A phishing game for phishing awareness) (P2: Strategies for counteracting social engineering attacks) etc.

Answer #2.3: We appreciate your comment and indeed the above work is useful when it comes to non technical methods for protection against BEC attacks and specifically in the section of employee training. For that reason, we added the reference (26) along with the informational text in the section of non technical methods.

Comment #2.4:  At the end of the introduction section, the authors are advised to add a flow diagram that guides the reader on how different sections are linked. This will help the reader understand the paper flow for better readability.

Answer #2.4: Thank you for highlighting the necessity to improve the readability of the paper. Following your advice, we added a flow diagram (Figure 2) that we think it guides the reader in how the different sections are linked. 

Comment #2.5: Figure 2 is taken from the source mentioned in the caption. Rather than using the same image, it would be better to refer to the same paper in the text.

Answer #2.5: Thank you for your comment. The paper in question is now referred in the text.

Comment #2.6: It seems that author's have written a report instead of paper. Authors are advised to use a template which is more readable to the reader. such as : Section 1: Introduction, Section 2: Literature Review, Section 3: Experiment Design, Section 4 : Results and Analysis, Section 5: Discussion and Conclusion. etc

Answer #2.6: Thank you for pointing this out. We added sections based on your suggestion in the revised paper that we hope to address this issue:

  • Contribution
  • Research Methodology
  • Results

Reviewer 3 Report

The paper is overall well written.

BEC attacks and corresponding defense strategies are thoroughly discussed.

Line 110 shows ???

Author Response

We would like to express our heartfelt gratitude for taking the time to review and provide feedback on our work. We truly appreciate your effort in helping me enhance the quality of our output. We have carefully considered each of your comments and suggestions, and we are pleased to present our replies to your comments below. 

Attached is our response, along with the revised version of our article, highlighting all the improved sections in yellow.

Comment #3.1: Comments on the Quality of English Language - Line 110 shows ???

Answer #3.1: Thank you for the review of our paper. You are totally right; Line 110 is an unintentionally mistake and it is corrected in the revised version. Thank you for pointing it out.

Reviewer 4 Report

The initial segment of this research paper presents a comprehensive review of Business Email Compromise (BEC) attacks, offering valuable recommendations to organizations aiming to bolster their cybersecurity defenses and mitigate the risks associated with such attacks. The authors emphasize the significance of combining non-technical and technical methods for effectively countering this type of fraud.

However, the scientific rigor employed in this approach is somewhat lacking, and the research's relevance needs further justification within the existing literature. It is crucial for the authors to include a methodology section to clarify the process of conducting the literature review. Is it a narrative literature review, a systematic review, or another type of review altogether? This information remains undisclosed, and it is essential to elucidate the methodology employed in the literature review process.

Moreover, the authors should explicitly address the limitations of their study to ensure transparency and provide a more comprehensive understanding of the research's scope and applicability. Additionally, the current Conclusion section appears to be stating the obvious and could benefit from further analysis and elaboration.

Furthermore, it is recommended that the authors expand the reference list to enhance the depth and breadth of the sources used to support their study.

As for the second part of the paper, which includes an analysis of the Greek cyberattack landscape, its relevance and importance to this study remain unclear. Consequently, it is advisable to remove this section to maintain the focus and coherence of the research.

none

Author Response

We would like to express our heartfelt gratitude for taking the time to review and provide feedback on our work. We truly appreciate your effort in helping me enhance the quality of our output. We have carefully considered each of your comments and suggestions, and we are pleased to present our replies to your comments below. 

Attached is our response, along with the revised version of our article, highlighting all the improved sections in yellow.

Comment #4.1: However, the scientific rigor employed in this approach is somewhat lacking, and the research's relevance needs further justification within the existing literature. It is crucial for the authors to include a methodology section to clarify the process of conducting the literature review. Is it a narrative literature review, a systematic review, or another type of review altogether? This information remains undisclosed, and it is essential to elucidate the methodology employed in the literature review process.

Answer #4.1: Thank you for your comment. We acknowledge the need for a research methodology and we added the section in the revised version (lines 97-134)

Comment #4.2: Moreover, the authors should explicitly address the limitations of their study to ensure transparency and provide a more comprehensive understanding of the research's scope and applicability. Additionally, the current Conclusion section appears to be stating the obvious and could benefit from further analysis and elaboration.

Answer #4.2: Thank you for your comment. We address the limitations of our research as suggested in Section Research Methodology as you may see in the revised version. We also added a Results Section and revised the discussion/ conclusion section to enhance its information quality as proposed.

Comment #4.3: Furthermore, it is recommended that the authors expand the reference list to enhance the depth and breadth of the sources used to support their study.

Answer #4.3: Thank you for bringing to our attention the importance of expanding the reference list. We have taken your suggestion into consideration and have included seven additional references in their respective sections, with the aim of enhancing the breadth and depth of our sources.

Comment #4.4: As for the second part of the paper, which includes an analysis of the Greek cyberattack landscape, its relevance and importance to this study remain unclear. Consequently, it is advisable to remove this section to maintain the focus and coherence of the research.

Answer #4.4: Thank you for your comment. We believe that it is informative to know the mechanics (departments, legislation) that a county like Greece has for dealing with cybercrime in general. Greece is listed in the top 20 countries by number of total victims as compared to the United States for 2022 while having the privilege of being a member of the European Union, thus making Greece a representative example from the authors point of view. The departments/ authorities listed in the article is more over the same in the countries across the EU, while we also mention the cooperative initiations that have risen in Europe to cope with cybercrime. That includes exchange of information and also Joined Investigation Teams, tools that are available only in the recent years.

             In the second part of this paper, we also report the procedures after a BEC attack and some steps to facilitate the work of the authorities that in our opinion is useful for someone unfamiliar with this type of scheme.

Finally, while the departments for dealing with cybercrime present a lot of similarities across the EU, the legislation adopted by the member states were lacking in legal tools and were sometimes outdated. The need for a universal legal framework arose along with the increase in loses due to malicious cyber activities and especially BEC attacks. For that reason, we believe it is important to state the legal measures that the European Union has taken (NIS 1, NIS 2, Directive 2022/ 2555 and ISO 27001/2022) which are implied to all state members. While Greece is given as an example, all measures and changes made from the NIS 1 and beyond have proven essential in protecting individuals, enterprises and critical infrastructures in all members states.   

             We acknowledge that we may have overlooked the general concept of how these two sections are interconnected in the Discussion and Conclusions section, as you mentioned in your previous comment. We apologize for any confusion caused. In order to rectify this, we have revised the aforementioned section to address this issue. We sincerely hope that our revised answer will meet your expectations and provide a satisfactory response.

Round 2

Reviewer 2 Report

The paper may be accepted.

Reviewer 4 Report

Thank you for taking the time to address the concerns and suggestions that were raised in the review of your work. I am pleased to see that you have provided detailed responses and made significant revisions to your article based on the comments provided.

The inclusion of additional explanations and justifications concerning the differences between your work and previously published work has enhanced the overall clarity and contribution of your study.